# *Alternaria alternata*, the Causal Agent of a New Needle Blight Disease on *Pinus bungeana*

**DOI:** 10.3390/jof9010071

**Published:** 2023-01-03

**Authors:** Mao-Jiao Zhang, Xiang-Rong Zheng, Huan Li, Feng-Mao Chen

**Affiliations:** 1College of Forestry, Nanjing Forestry University, Nanjing 210037, China; 2Collaborative Innovation Center of Sustainable Forestry in Southern China, Nanjing 210037, China; 3College of Landscape Architecture, Jiangsu Vocational College of Agriculture and Forestry, Zhenjiang 212400, China

**Keywords:** *Pinus bungeana*, *Alternaria alternata*, needle blight, identification

## Abstract

*Pinus bungeana*, an endangered and native coniferous tree species in China, has considerable timber and horticulture value. However, little is known about needle diseases in *P. bungeana*. A needle blight of *P. bungeana* has been observed in Hebei Province, China. *P. bungeana* inoculated with mycelial plugs of fungal isolates presented symptoms similar to those observed under field conditions. Ten virulent fungal isolates were identified as a small-spored *Alternaria* species based on morphological observations. Maximum likelihood and Bayesian phylogenetic analyses carried out with multilocus sequence typing of eight regions (SSU, LSU, ITS, *gapdh*, *tef1*, *Alt a 1*, *endoPG*, OPA10-2) assigned the pathogen to *Alternaria alternata*. This is the first report of *A. alternata* causing needle blight on *P. bungeana* in China.

## 1. Introduction

Bunge’s pine (*Pinus bungeana* Zucc. ex Endl.), a distinctive and evergreen coniferous tree species within the genus *Pinus* of the family Pinaceae, is mainly distributed across warm temperate areas and the north-subtropical and middle subtropical climatic zones [1]. It is known as an endemic and endangered coniferous tree species in China with high ornamental value, and is widely used in landscaping and afforestation owing to its ability to endure drought and cold climates [2]. Furthermore, the wood of Bunge’s pine is commonly used for construction, furniture and stationery [3]. In addition, *P. bungeana* plays a key role in local forest ecosystems, with strong resistance to sulphur dioxide, ozone and soot pollution in nature [4]. Due to its ecological and economic value, this species has been the subject of many investigations, mainly on its phylogeny, morphology, genetic diversity and biological characteristics [5,6]. Few diseases of Bunge’s pine have been reported.

The genus *Alternaria* Nees was described in 1816 [7]. Since then, more than 1100 names have been published, and 275 *Alternaria* species have been recognised [8,9]. *Alternaria* is a ubiquitous fungal genus that includes saprophytic, endophytic and pathogenic species [10]. Some *Alternaria* species are famous as pathogens of plants and animals [11]. In addition, those pathogenic species harm more than 4000 host plants and are distributed worldwide, with a broad host range, including agronomic plants, ornamentals, vegetables, fruit trees and animals [10,12,13]. Leaf blight, leaf spot, black point, stem cancer, fruit rot and mouldy cores are well-known symptoms of infection by *Alternaria* species [14,15,16].

In the past, *Alternaria* spp. have been classified based exclusively upon their morphological characteristics, which include cultural morphology, shape and size of conidia, septation, beak formation, branching patterns of conidial chains, and sporulation patterns [17]. This approach is effective when distinguishing large-spored *Alternaria* spp. from small-spored catenulate species due to conidia that are distinct and easy to recognise. Nevertheless, the identification of small-spored species based on morphological characteristics is challenging due to the overlap of many morphological traits [18]. Therefore, using different molecular tools to support morphological inference for *Alternaria* taxonomy is essential. These tools include DNA fingerprinting techniques (RAPD, PCR-RFLP, AFLP and ISSR) and sequence analysis of rDNA and protein coding genes, such as nuclear internal transcribed-spacer regions (ITS), the mitochondrial ribosomal large subunit (mtLSU), the mitochondrial small subunit (mtSSU), translation elongation factor (TEF), beta-tubulin, endopolygalacturonase (*endoPG*) genes, glyceraldehyde-3-phosphate dehydrogenase (GAPDH) gene, RNA polymerase second largest subunit (RPB2), plasma membrane ATPase, Alternaria major allergen gene (*Alt a 1*), calmodulin (CAL) and the anonymous genomic regions OPA1-3, OPA2-1 and OPA10-2 [7,18,19,20,21,22,23]. Among these genes, the plasma membrane ATPase and calmodulin loci were proposed as the most suitable genetic markers for the molecular identification of small-spored *Alternaria* [7,11]. Furthermore, the histone 3 gene (HIS3) has been used to separate *A. alternata* from *A. tenuissima* [24,25].

*Alternaria alternata* (Fr.) Keissl, the type species for the genus *Alternaria*, is able to cause diseases in over 100 plants, including vegetables, fruits, herbs and ornamental trees [26,27,28,29]. Additionally, it can cause postharvest disease in various crops and respiratory diseases in humans [20]. It is a causal agent that gives rise to leaf spot, leaf blight and mouldy cores in host plants [15,30,31]. In addition, a serious infection risk was posed to horticultural crops all over the world because of the rapid market globalization of the seeds, long-distance airborne transmission of spores and the influences of changed climate [10].

A few diseases related to *P. bungeana* have been reported, including needle cast, trunk rot, needle rusts and twig blight [32,33]. However, there are no reports about needle blight in *P. bungeana.* The aim of this study was to identify the pathogens that cause needle blight in Bunge’s pine using morphological and molecular phylogenetic approaches and lay a theoretical foundation for the control of this pathogen.

## 2. Materials and Methods

### 2.1. Disease Investigation and Isolate Collection

In September 2020, leaf spot of Bunge’s pine was found in Hebei Province, China. Thirty symptomatic tissues, the margin between the lesioned and healthy pine leaves, were cut into 3 to 5 mm long pieces. These tissues were surface sterilised for 45 s in ethanol (75%), washed thrice in sterilised distilled water and blotted dry with sterile paper. Pieces were transferred to 2% potato dextrose agar (PDA) in Petri plates, supplemented with ampicillin at 100 μg/mL and incubated at 25 °C (±1 °C) in the dark for 4 days. The single-spore isolation technique was used to obtain purified fungal isolates [34]. Single-spore isolates were cultured on PDA and stored in the Forest Pathology Laboratory of Nanjing Forestry University, Nanjing, China and the representative strain are being deposited to China Center for Type Culture Collection, Wuhan, China (CCTCC).

### 2.2. Pathogenicity Tests

All isolates were cultured on PDA and used for virulence tests on detached *P. bungeana* needles under controlled conditions. Asymptomatic needles of *P. bungeana* were surface disinfected and air-dried. Then, one piercing wound was made on the mid-upper region of each needle with a sterile needle (insect pin, 0.71 mm in diameter). The inoculation was performed by placing mycelial blocks (5 mm in length) from actively growing colony margins onto each stab wound. Needles inoculated with noncolonised PDA blocks were treated as negative controls. Each control and treatment, involving three needles per replicate, was placed into a Petri dish (9 cm) with moist sterile filter paper and sealed with plastic wrap to maintain a high relative humidity. Then, they were incubated at 25 °C in a growth chamber with a 12 h photoperiod. The whole experiment was carried out three times.

Ten isolates that were confirmed to be pathogenic on the detached needles were selected to determine pathogenicity on potted Bunge’s pine. Bunge’s pine needles were disinfected with 75% ethanol and air dried. Then, wound inoculation was conducted on 2-year-old potted, healthy Bunge’s pine with a sterile needle. The blocks (3 mm in length) from colony margins with actively growing mycelia of 3-day-old isolates were placed on each wounded site. Blocks were removed 2 days post-inoculation. PDA discs with no mycelia were used as controls. Three potted plants were treated as one replicate, and three replicates were used. The inoculated plants were placed into a controlled-environment greenhouse. The size of the disease spot was recorded until representative symptoms appeared. The same procedure was carried out on 2-month-old seedlings of Korean pine (*P. koraiensis* Sieb. et Zucc.).

Re-isolations were performed from the margins of needles inoculated with ten isolates, and morphological and phylogenetic comparisons were conducted to meet Koch’s postulates.

### 2.3. Morphological Study

Isolates were cultured on PDA for 7 days at 25 °C (±1 °C) to observe the colony morphology [35]. Micromorphological features were observed from those cultured on synthetic nutrient-poor agar plates (SNA) [36]. The characteristics of sporulation formation, including the length of conidial chains, branching patterns of conidial chains and presence of secondary conidiophores, were captured with a Zeiss stereo microscope (SteRo Discovery v20) [35]. A ZEISS Axio Imager A2m microscope (Carl Zeiss, Göttingen, Germany) equipped with differential interference contrast (DIC) optics was used to capture conidial chains and conidia. Fifty mature conidia mounted in sterile water were measured at random under a light microscope at ×100 magnification.

### 2.4. DNA Extraction and Polymerase Chain Reaction (PCR)

The CTAB method described by Damm et al. [37] was used to extract genomic DNA from isolates that had been cultured on PDA at 25 °C for 5 days. The ITS, *tef1*, *endoPG*, OPA10-2, *Alt a1*, SSU, LSU and *gapdh* genes were amplified with the primer pairs V9G/ITS4 [38,39], EF1-728F/EF1-986R [40], PG3/PG2b [18], OPA10-2L/OPA10-2R [18], Alt-for/Alt-rev [41], NS1/NS4 [39], LSU1Fd/LR5 [42,43], and gpd1/gpd2 [44], respectively. Polymerase chain reaction (PCR) amplification was conducted in a total reaction volume of 25 μL containing 12.5 μL Taq DNA solution, 1 μL of each primer (10 pmol/μL), 2 μL (100 ng) of genomic DNA and 8.5 µL of double-distilled H_2_O with a thermal cycler under the conditions listed in Table 1. The PCR products were electrophoresed (160 V for 20 min) on 2% agarose gels and sequenced bidirectionally at the Shanghai Sangon Biological Technology Company (Shanghai, China) using Sanger DNA Sequencing from both directions. The sequenced DNA products were deposited at the National Centre for Biotechnology Information (NCBI) (Table 2).

### 2.5. DNA Sequencing and Phylogenetic Analysis

The reference sequences of 43 *Alternaria* spp. described by Woudenberg et al. [20] selected for the phylogenetic analyses are also listed in Table 2 together with their corresponding GenBank accession numbers. Sequences in Table 2 were retrieved from the GenBank database (https://www.ncbi.nlm.nih.gov/ (accessed on 10 February 2021)). *A. alternantherae* (CBS 124392) was used as the outgroup. The alignments of nucleotide sequences were obtained by using Clustal W in BioEdit software [45]. Treating gaps in the alignment as a fifth character, all of the characters had equal weight [46].

Phylogenetic trees of combined genes were constructed with two independent optimality search criteria, Bayesian inference (BI) phylogenetic analysis and maximum likelihood (ML) analysis. The ML analysis was performed using IQ-TREE [47], choosing the GTR + G + I model, and branch stability was estimated by 1000 bootstrap replicates. The BI analysis was performed in PhyloSuite version 1.2.2. using Mr. Bayes v. 3.2.6. [48] under a partition model (2 parallel runs, 1 × 10^7^ generations), with FigTREE v1.4.4 used to view the phylogenetic trees.

## 3. Results

### 3.1. Symptoms in Nature

Symptoms appeared on Bunge’s pine needles and enlarged constantly. The colour of infected needles is off-white at the early stage and then turns to light brown gradually, with dark-brown spots appearing one by one (Figure 1B,C). At the later stage of the disease, a large number of needles are infected, and the growth of the tree is inhibited (Figure 1A). In total, 20 single-spore fungal isolates were collected.

### 3.2. Pathogenicity Tests

Ten isolates were pathogenic, and healthy needles exhibited symptoms similar to those in nature, while mock-inoculated control needles showed no symptoms (Figure 2). Light-brown lesions were first observed at two days after inoculation and then expanded gradually, and dark-brown segments were noticed 14 days after mycelial plug inoculation (Figure 2B). Ten lesions of each strain were counted and there was no significant difference in virulence among the three strains. Symptoms in nature appeared on Korean pine seedlings (Figure 2C). The fungus was re-isolated from inoculated needles, and its colony morphology and molecular sequence were consistent with those of the original isolates.

### 3.3. Morphology of Fungal Isolates

The virulent isolates shared similar colony morphologies. The colonies, with a regular prominent white margin, were olive green to black 10 d post-incubation. The bottom of the colonies was black surrounded with a light-brown circle. The aerial hyphae were thick and cottony and turned from colourless to pale brown (Figure 3A). Conidiophores arose singly and were separated and pale brown. The conidia were solitary or in chains, the conidial body was 18.09–37.61 μm × 9.15–19.90 μm (average 24 × 14 μm, *n* = 50), typically obclavate, subglobose and ellipsoid, with 1–5 transverse septa and 1–3 longitudinal septa that slightly constricted near several septa. The conidia were yellow–brown and later turned black–brown (Figure 3B–F). The morphological characterization of ten isolates revealed *Alternaria*-like morphology.

### 3.4. Phylogenetic Analysis

A multilocus phylogenetic analysis was conducted on ten pathogenic isolates based on the sequences from eight genes: SSU, LSU, ITS, *gapdh*, *tef1*, *Alt a 1*, *endoPG* and OPA10-2 (GenBank accession numbers MZ835355 to MZ835364, MZ835345 to MZ835354, MZ823461 to MZ823470, MZ835385 to MZ835394, MZ835395 to MZ835404, MZ802959 to MZ802968, MZ835375 to MZ835384, MZ835365 to MZ835374). For these ten isolates, the PCR amplification and sequencing of each gene generated product sizes were about 1072, 942, 733, 619, 259, 516, 491 and 753 or 777 bp, respectively. The alignments (including the gaps) for eight genes were 1021, 849, 522, 579, 241, 473, 448 and 634 bp in size, respectively. The ten sequences of isolates along with sequences from 33 Alternaria strains were concatenated for the construction of a phylogenetic tree. The alignment of the eight-locus concatenated dataset consisted of 4767 characters, with 4356 constant characters, 245 parsimony-uninformative characters, and 166 parsimony-informative characters.

ML and BI analyses generated basically the same tree topology, which demonstrated that the evolutionary relationships of the fungus isolates were statistically supported. A single tree with bootstrap proportions (BP) from ML and Bayesian posterior probabilities (BPP) from BI was generated (Figure 4). The phylogenetic analysis showed that all isolates herein clustered into two clades, with a highly supported clade (≥92% BP/0.91 BPP) with *A. alternata* CBS 121455 and CBS 121336. Two phylogenetic analyses revealed that all isolates with aggressiveness showed >95% similarity to the *A. alternata* isolates reported previously.

## 4. Discussion

Because of its ability to assimilate harmful material in the needles, graceful appearance and fine timber, Bunge’s pine plays an essential role in ecology and the economy. Needle blight disease can not only worsen the pine appearance but also influence apical dominance. The loss of apical dominance reduces wood quality. Moreover, the death of trees can occur in severe cases. Generally, the diseases affecting Bunge’s pine damage the economy and ecology. Based on morphological characteristics and molecular identification with phylogenetic analysis of multiple gene sequences, *A. alternata* was confirmed to be the causal agent of needle blight on Bunge’s pine in China. This is the first report of *A. alternata* on *P. bungeana*.

Several small-spore *Alternaria* spp. are frequently misidentified due to morphological overlap with *A. alternata* [35]. The dimensions of conidia in this study were very different from those described by Moumni et al. [49], but were similar to those reported by Gao et al. [50]. This phenomenon could be attributed to the morphological plasticity exhibited by most *Alternaria* species. Conidial morphology is dependent on culture conditions and conidium age [35]. The number of conidia produced with conidial chains was related to the nutrition that the fungi obtained. In addition, the numbers of longitudinal and transverse septa were variable. It is suggested that morphological characteristics are not stable.

Due to morphological variability and minimal molecular variation, the taxa of *Alternaria* spp. were reclassified by Woundenberg et al. [51]. Whole-genome sequencing and transcriptome sequencing were used to distinguish 168 *Alternaria* isolates, and nine gene regions (SSU, LSU, ITS, *gapdh*, *tef1*, *Alt a 1*, *endoPG*, OPA10-*2* and *rpb2*) were selected to distinguish sect. *Alternaria* more effectively [20]. Phylogenetic analyses and species identification are challenging in small-spored *Alternaria* due to lineage sorting, recombination and horizontal transfer [52]. Multilocus species identification was confirmed to be necessary among *Alternaria* sections for low resolution of species delimitation in small-spored *Alternaria* [10]. The analysis with a concatenation of six gene regions (ITS, *rpb2*, *endoPG*, *tef1*, *Alt a 1* and OPA10-2) was able to separate *A. alternata* from the *A. arborescens* species complex [10]. A slowly evolving gene (*rpb2*) was excluded, while additional molecular markers (*gaphd*, SSU and LSU) were included in this study as proposed by Woudenberg et al. [20]. The combined phylogenetic tree shows consistency with other studies [10,15,17,20].

*Alternaria alternata* was reported as a ubiquitous pathogen in the great majority of crops and some broad-leaved trees [17,26,30,31,53,54,55,56]. In particular, *A. alternata* is the most important mycotoxin-producing genus as a result of the wide reports of TA, AME, AOH, ALT and ATX produced [57]. In addition, *A. alternata* can not only colonise the phylloplane but also penetrate into living leaves [58]. Nevertheless, *A. alternata* was reported to be the dominant endophytic fungal taxon in the bark and needles of Chinese oil pine (*Pinus tabulaeformis* Carr.) and isolated from various plants [59]. In addition, as an endophytic fungus, it showed strong antifungal activity against *Raffaelea quercus-mongolicae* [60]. When examining the abundance and diversity of fungi on needles of *Pinus sylvestris*, *A. alternata* was found to be a common primary or secondary saprotroph [61]. It is difficult for *A. alternata* to colonise Bunge’s pine needles without wounding, which may be related to plant resistance or pathogenic activity. The result of unwounded inoculation indicated that wounding may play a significant role in the pathogenicity of *A. alternata.* In nature, needles are prone to chafing, which can induce laceration as a result of the wind. This may provide an opportunity for *A. alternata* to be virulent. In addition, the virulence of *A. alternata* may have been obtained horizontally from a recent common saprophytic ancestor [52].

According to previous studies, *A. alternata*, as a pathogen of pine needles, has never been reported. Although the thicker epidermis and cuticle of needles make it more difficult for fungi to invade plants, it is noteworthy that wounds appearing on needles may lead to disease prevalence. Pathogenicity test results indicate that *A. alternata* has the ability to infect other *Pinus* species, and it is necessary to investigate the distribution and propagation of the disease caused by *A. alternata*. *A. alternata* may pose a great threat to ecology because the hosts that the pathogen can invade are increasing, especially in *Pinus* species. Studies on the pathogenicity mechanism of *A. alternata* and disease management should be conducted in the future.

## Figures and Tables

**Figure 1 jof-09-00071-f001:**
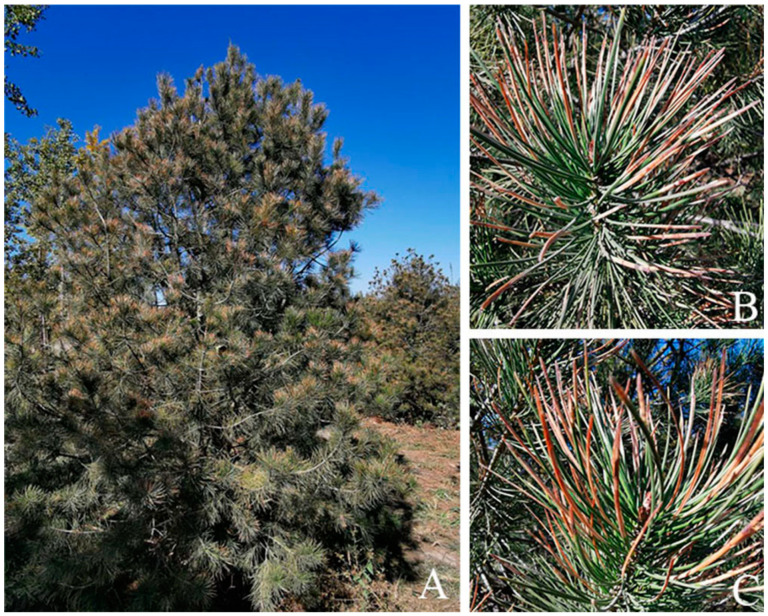
Symptoms of infection by *A. alternata* on *P. bungeana* in the field. (**A**), Withered tips of the whole tree. (**B**,**C**), Magnified image showing symptoms on needles.

**Figure 2 jof-09-00071-f002:**
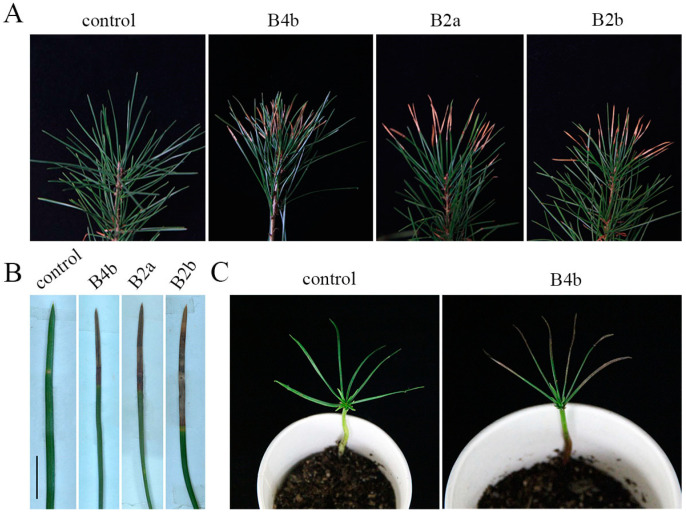
Pathogenicity of *A. alternata* on Bunge’s pine and Korean pine achieved by mycelial discs. (**A**) Pathogenicity on 2-year-old seedlings of Bunge’s pine. (**B**) Pathogenicity on detached needles of Bunge’s pine. (**C**) Pathogenicity on 2-month-old seedlings of Korean pine. Scale bars: (**B**) = 5 mm.

**Figure 3 jof-09-00071-f003:**
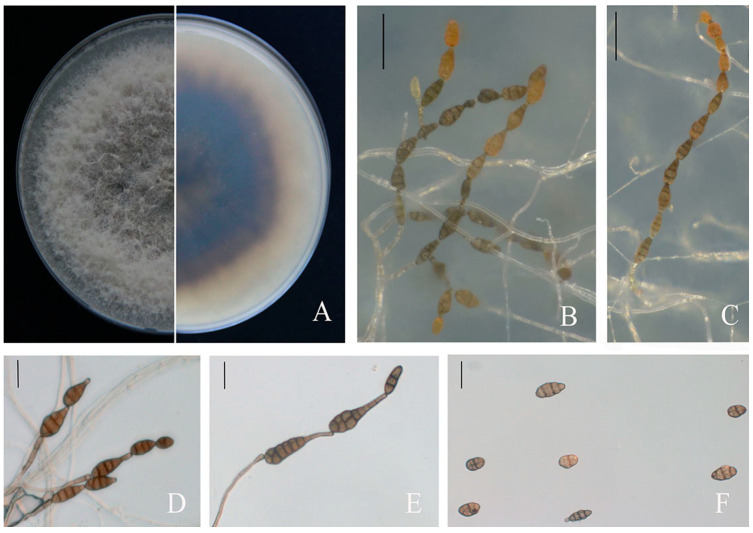
Morphological characters of *A. alternata.* (**A**) Front and back views of colony morphology on PDA after 7 days. (**B**–**E**) Conidiophores developed on SNA. (**F**) Conidia. Scale bars: (**B**,**C**) = 50 µm; (**D**–**F**) = 20 µm.

**Figure 4 jof-09-00071-f004:**
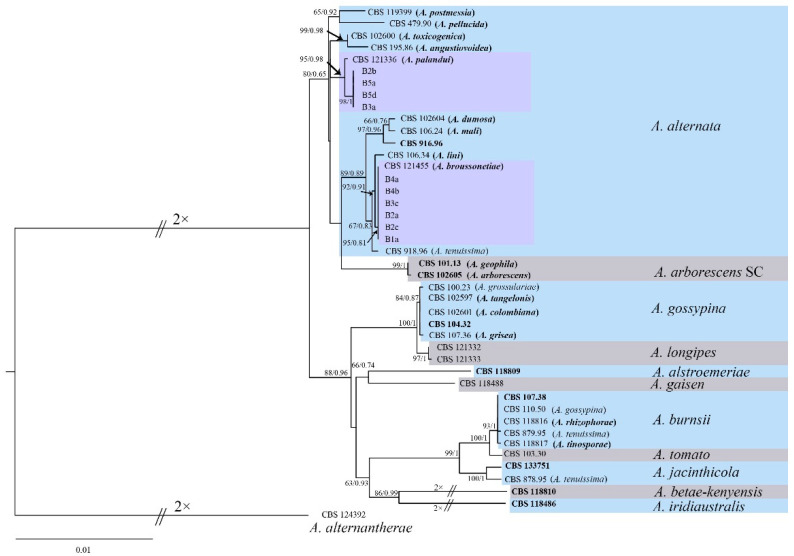
Maximum likelihood and Bayesian analyses of 43 isolates of the *Alternaria* species. The tree was generated with concatenated sequences of the SSU, LSU, ITS, *gapdh*, *tef1*, *Alt a 1*, *endoPG* and OPA10-2 regions or genes. The tree generated by Bayesian inference had a similar topology. Bootstrap support values above 60% (before the slash marks) and Bayesian posterior probability values above 0.75 (after the slash marks) are shown at each node. Species names in parentheses refer to the former species name. Ex-type strains are emphasised in bold. *A. alternantherae* CBS 124392 was used as an outgroup. The scale bar shows the predicted number of substitutions per nucleotide position.

**Table 1 jof-09-00071-t001:** List of the primers used for PCR and sequencing.

Locus	Primer	Sequence (5′-3′)	PCR Conditions	Reference
Internal transcribed spacer (ITS)	V9G	TTACGTCCCTGCCCTTTGTA	Denaturation for 3 min at 94 °C, followed by 30 cycles; 30 s at 94 °C, 30 s at 48 °C, 30 s at 72 °C, and 10 min of a final extension at 72 °C	[38]
	ITS4	TCCTCCGCTTATTGATATGC		[39]
Elongation factor 1-alpha (*tef1*)	EF1-728F	CATCGAGAAGTTCGAGAAGG	Denaturation for 3 min at 94 °C, followed by 30 cycles; 30 s at 94 °C, 30 s at 55 °C, 30 s at 72 °C, and 10 min of a final extension at 72 °C	[40]
	EF1-986R	TACTTGAAGGAACCCTTACC		
Endopolygalacturonase (*endoPG*)	PG3	TACCATGGTTCTTTCCGA	Denaturation for 3 min at 94 °C, followed by 30 cycles; 30 s at 94 °C, 30 s at 50 °C, 30 s at 72 °C, and 10 min of a final extension at 72 °C	[18]
	PG2b	GAGAATTCRCARTCRTCYTGRTT		
Anonymous gene region (OPA 10-2)	OPA 10-2R	GATTCGCAGCAGGGAAACTA	Denaturation for 3 min at 94 °C, followed by 30 cycles; 30 s at 94 °C, 30 s at 58 °C, 30 s at 72 °C, and 10 min of a final extension at 72 °C	[18]
	OPA 10-2L	TCGCAGTAAGACACA TTCTACG		
Alternaria major allergen gene (*Alt a 1*)	Alt-for	ATGCAGTTCACCACCATCGC	Denaturation for 3 min at 94 °C, followed by 30 cycles; 30 s at 94 °C, 30 s at 60 °C, 30 s at 72 °C, and 10 min of a final extension at 72 °C	[41]
	Alt-rev	ACGAGGGTGAY GTAGGCGTC		
18S nrDNA (SSU)	NS1	GTAGTCATATGCTTGTCTC	Denaturation for 3 min at 94 °C, followed by 30 cycles; 30 s at 94 °C, 30 s at 55 °C, 30 s at 72 °C, and 10 min of a final extension at 72 °C	[39]
	NS4	CTTCCGTCAATTCCTTTAAG		
28S nrDNA (LSU)	LSU1Fd	GRATCAGGTAGG RATACCCG	Denaturation for 3 min at 94 °C, followed by 30 cycles; 30 s at 94 °C, 30 s at 51 °C, 30 s at 72 °C, and 10 min of a final extension at 72 °C	[42]
	LR5	ATCCTGAGGGAAACTTC		[43]
glyceraldehyde-3-phosphate dehydrogenase (*gapdh*)	gpd1	CAACGGCTTCGGTCG CATTG	Denaturation for 3 min at 94 °C, followed by 30 cycles; 30 s at 94 °C, 30 s at 57 °C, 30 s at 72 °C, and 10 min of a final extension at 72 °C	[44]
	gpd2	GCCAAGCAGTTGGTTGTGC		

**Table 2 jof-09-00071-t002:** Descriptions and sequence accession numbers obtained from GenBank of *Alternaria* spp. used in the phylogenetic study.

Species Name and Strain Number ^1,2^	Locality, Host/Substrate	GenBank Accession Numbers ^3^							
		SSU	LSU	ITS	*gapdh*	*tef1*	*Alt a 1*	*endoPG*	OPA10-2
*Alternaria alstroemeriae*									
CBS 118809 ^T^	Australia, *Alstroemeria* sp.	KP124918	KP124448	KP124297	KP124154	KP125072	np	KP123994	KP124602
*Alternaria alternantherae*									
CBS 124392	China, *Solanum melongena*	KC584506	KC584251	KC584179	KC584096	KC584633	KP123846	np	np
*Alternaria alternata*									
B4b	China, *Pinus bungeana*	**MZ835355**	**MZ835345**	**MZ823461**	**MZ835385**	**MZ835395**	**MZ802959**	**MZ835375**	**MZ835365**
B2c	China, *Pinus bungeana*	**MZ835356**	**MZ835346**	**MZ823462**	**MZ835386**	**MZ835396**	**MZ802960**	**MZ835376**	**MZ835366**
B4a	China, *Pinus bungeana*	**MZ835357**	**MZ835347**	**MZ823463**	**MZ835387**	**MZ835397**	**MZ802961**	**MZ835377**	**MZ835367**
B1a	China, *Pinus bungeana*	**MZ835358**	**MZ835348**	**MZ823464**	**MZ835388**	**MZ835398**	**MZ802962**	**MZ835378**	**MZ835368**
B2a	China, *Pinus bungeana*	**MZ835359**	**MZ835349**	**MZ823465**	**MZ835389**	**MZ835399**	**MZ802963**	**MZ835379**	**MZ835369**
B3c	China, *Pinus bungeana*	**MZ835360**	**MZ835350**	**MZ823466**	**MZ835390**	**MZ835400**	**MZ802964**	**MZ835380**	**MZ835370**
B2b	China, *Pinus bungeana*	**MZ835361**	**MZ835351**	**MZ823467**	**MZ835391**	**MZ835401**	**MZ802965**	**MZ835381**	**MZ835371**
B5d	China, *Pinus bungeana*	**MZ835362**	**MZ835352**	**MZ823468**	**MZ835392**	**MZ835402**	**MZ802966**	**MZ835382**	**MZ835372**
B5a	China, *Pinus bungeana*	**MZ835363**	**MZ835353**	**MZ823469**	**MZ835393**	**MZ835403**	**MZ802967**	**MZ835383**	**MZ835373**
B3a	China, *Pinus bungeana*	**MZ835364**	**MZ835354**	**MZ823470**	**MZ835394**	**MZ835404**	**MZ802968**	**MZ835384**	**MZ835374**
CBS 916.96 ^T^	India, *Arachis hypogaea*	KC584507	DQ678082	AF347031	AY278808	KC584634	AY563301	JQ811978	KP124632
CBS 195.86 (*A. angustiovoidea* ^T^)	Canada, *Euphorbia esula*	KP124939	KP124469	KP124317	KP124173	KP125093	JQ646398	KP124017	KP124624
CBS 106.24 (*A. mali* ^T^)	USA, *Malus sylvestris*	KP124919	KP124449	KP124298	KP124155	KP125073	KP123847	AY295020	JQ800620
CBS 102604 (*A. Dumosa* ^T^)	Israel, *Minneola tangelo*	KP124956	KP124486	KP124334	AY562410	KP125110	AY563305	KP124035	KP124643
CBS 106.34 (*A. lini* ^T^)	Unknown, *Linum usitatissimum*	KP124924	KP124454	Y17071	JQ646308	KP125078	KP123853	KP124000	KP124608
CBS 918.96 (*A. tenuissima* ^R^)	UK, *Dianthus chinensis*	KC584567	KC584311	AF347032	AY278809	KC584693	AY563302	KP124026	KP124633
CBS 479.90 (*A. pellucid*a ^T^)	Japan, *Citrus unshiu*	KP124941	KP124471	KP124319	KP124174	KP125095	KP123870	KP124019	KP124626
CBS 102600 (*A. toxicogenica* ^T^)	USA, *Citrus reticulata*	KP124953	KP124483	KP124331	KP124186	KP125107	KP123880	KP124033	KP124640
CBS 119399 (*A. postmessia* ^T^)	USA, *Minneola tangelo*	KP124983	KP124513	KP124361	JQ646328	KP125137	KP123910	KP124063	KP124672
CBS 121336 (*A. palandui* ^T^)	USA, *Allium* sp.	KP124987	KP124517	KJ862254	KJ862255	KP125141	KJ862259	KP124067	KP124676
CBS 121455 (*A. broussonetiae* ^T^)	China, *Broussonetia papyrifera*	KP124992	KP124522	KP124368	KP124220	KP125146	KP123916	KP124072	KP124681
*Alternaria arborescens* species complex (AASC)									
CBS 101.13 (*A. geophila* ^T^)	Switzerland, peat soil	KP125016	KP124546	KP124392	KP124244	KP125170	KP123940	KP124096	KP124705
CBS 102605 (*A. arborescens* ^T^)	USA, *Solanum lycopersicum*	KC584509	KC584253	AF347033	AY278810	KC584636	AY563303	AY295028	KP124712
*Alternaria betae-kenyensis*									
CBS 118810 ^T^	Kenya, *Beta vulgaris* var. *cicla*	KP125042	KP124572	KP124419	KP124270	KP125197	KP123966	KP124123	KP124733
*Alternaria burnsii*									
CBS 107.38 ^T^	India, *Cuminum cyminum*	KP125043	KP124573	KP124420	JQ646305	KP125198	KP123967	KP124124	KP124734
CBS 110.50 (*A. gossypina*)	Mozambique, *Gossypium* sp.	KP125044	KP124574	KP124421	KP124271	KP125199	KP123968	KP124125	KP124735
CBS 879.95 (*A. tenuissima*)	UK, *Sorghum* sp.	KP125045	KP124575	KP124422	KP124272	KP125200	KP123969	KP124126	KP124736
CBS 118816 (*A. rhizophorae* ^T^)	India, *Rhizophora mucronata*	KP125046	KP124576	KP124423	KP124273	KP125201	KP123970	KP124127	KP124737
CBS 118817 (*A. tinosporae* ^T^)	India, *Tinospora cordifolia*	KP125047	KP124577	KP124424	KP124274	KP125202	KP123971	KP124128	KP124738
*Alternaria gaisen*									
CBS 118488 ^R^	Japan, *Pyrus pyrifolia*	KP125051	KP124581	KP124427	KP124278	KP125206	KP123975	KP124132	KP124743
*Alternaria gossypina*									
CBS 100.23 (*A. grossulariae*)	Unknown, *Malus domestica*	KP125053	KP124583	KP124429	KP124280	KP125208	KP123977	KP124134	KP124745
CBS 104.32 ^T^	Zimbabwe, *Gossypium* sp.	KP125054	KP124584	KP124430	JQ646312	KP125209	JQ646395	KP124135	KP124746
CBS 107.36 (*A. grisea* ^T^)	Indonesia, soil	KP125055	KP124585	KP124431	JQ646310	KP125210	JQ646393	KP124136	KP124747
CBS 102597 (*A. tangelonis* ^T^)	USA, *Minneola tangelo*	KP125056	KP124586	KP124432	KP124281	KP125211	KP123978	KP124137	KP124748
CBS 102601 (*A. colombiana* ^T^)	Colombia, *Minneola tangelo*	KP125057	KP124587	KP124433	KP124282	KP125212	KP123979	KP124138	KP124749
*Alternaria iridiaustralis*									
CBS 118486 ^T^	Australia, *Iris* sp.	KP125059	KP124589	KP124435	KP124284	KP125214	KP123981	KP124140	KP124751
*Alternaria jacinthicola*									
CBS 878.95 (*A. tenuissima*)	Mauritius, *Arachis hypogaea*	KP125061	KP124591	KP124437	KP124286	KP125216	KP123983	KP124142	KP124753
CBS 133751 ^T^	Mali, *Eichhornia crassipes*	KP125062	KP124592	KP124438	KP124287	KP125217	KP123984	KP124143	KP124754
*Alternaria longipes*									
CBS 121333 ^R^	USA, *Nicotiana tabacum*	KP125068	KP124598	KP124444	KP124293	KP125223	KP123990	KP124150	KP124761
CBS 12133	USA, *Nicotiana tabacum*	KP125067	KP124597	KP124443	KP124292	KP125222	KP123989	KP124149	KP124760
*Alternaria tomato*									
CBS 103.30	Unknown, *Solanum lycopersicum*	KP125069	KP124599	KP124445	KP124294	KP125224	KP123991	KP124151	KP124762

^1^ CBS: Culture collection of the Centraalbureau voor Schimmelcultures, Fungal Biodiversity Centre, Utrecht, The Netherlands. ^2 T^: ex-type isolate; ^R^: representative isolate; Species names in parentheses refer to the former species name. ^3^ Bold accession numbers were generated in this study; np: no product.

## Data Availability

All data generated or analyzed during this study are included in this article.

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
