# Peer review of "Alternaria alternata, the Causal Agent of a New Needle Blight Disease on Pinus bungeana"

_jof, 2023, doi:10.3390/jof9010071_

Round 1

Reviewer 1 Report

This is an excellently written paper describing a new pathogen of a fascinating, important and beautiful tree.

I found nothing wrong with this paper and recommend acceptance either as is or with very minor revisions. I light edit for English usage would strengthen it even more.

Author Response

Thank you very much for your approval and spending your precious time reviewing the manuscript.

Reviewer 2 Report

Dear authors, 

The manuscript describing "Alternaria alternata, the causal agent of a new needle blight disease on Pinus bungeana" is well written and provides novel information. 

I have a few concerns as below

1. Table 1 can you provide the reference to each marker, rather than in text, it's better for the reading. 

2. line no 128-130 Pls mention the amplicon size that sequenced, the method of the sequencing   

3. In the phylogenic tree, can you highlight your isolates in a different colour it would be good to identify. 

4. Being the first report, have you deposited your isolates culture to any IDA microbial repository? If not pls, deposit and provide the accession number. 

5. Table A1 can be part of the main text of the manuscript. so pls include it. 

The paper needs minor revisions before acceptance.

All the best

Author Response

Response to Reviewer 2 Comments

Point 1: Table 1 can you provide the reference to each marker, rather than in text, it's better for the reading. 

Response 1: We have added a list of reference which corresponding to each primer to the table.

Point 2: line no 128-130 Pls mention the amplicon size that sequenced, the method of the sequencing.

Response 2: The method of sequencing is Sanger DNA Sequencing that have been added to line 130 and the amplicon size that sequenced was added in Results 3.4 line 194-196.

Point 3: In the phylogenic tree, can you highlight your isolates in a different colour it would be good to identify. 

Response 3: We have highlighted the isolate in different colour and revised Figure 4 in the manuscript.

Point 4: Being the first report, have you deposited your isolates culture to any IDA microbial repository? If not pls, deposit and provide the accession number. 

Response 4: At present, the epidemic caused by Corona Virus Disease2019 (COVID-19) brings inconvenience to express delivery transportation in China. It’s difficult for us to deposited the isolate culture to any IDA microbial repository. In addition, we attach great importance to the preservation of strains and there’s a special place for storing isolates culture in the Forest Pathology Laboratory of Nanjing Forestry University.

Point 5: Table A1 can be part of the main text of the manuscript. so pls include it.

Response 5: We have change Table A1 to Table 2 and adjusted its position to the first time mentioned in the manuscript.